# Lipid-Independent Regulation of PLIN5 via IL-6 through the JAK/STAT3 Axis in Hep3B Cells

**DOI:** 10.3390/ijms24087219

**Published:** 2023-04-13

**Authors:** Marinela Krizanac, Paola Berenice Mass Sanchez, Sarah K. Schröder, Ralf Weiskirchen, Anastasia Asimakopoulos

**Affiliations:** Institute of Molecular Pathobiochemistry, Experimental Gene Therapy and Clinical Chemistry (IFMPEGKC), RWTH University Hospital Aachen, D-52074 Aachen, Germany

**Keywords:** perilipin, fat metabolism, NAFLD, NASH, HCC, cytokines, IL-6, JAK/STAT3, transforming growth factor-β, tumor necrosis factor-α

## Abstract

Perilipin 5 (PLIN5) is a lipid droplet coat protein that is highly expressed in oxidative tissues such as those of muscles, the heart and the liver. PLIN5 expression is regulated by a family of peroxisome proliferator-activated receptors (PPARs) and modulated by the cellular lipid status. So far, research has focused on the role of PLIN5 in the context of non-alcoholic fatty liver disease (NAFLD) and specifically in lipid droplet formation and lipolysis, where PLIN5 serves as a regulator of lipid metabolism. In addition, there are only limited studies connecting PLIN5 to hepatocellular carcinoma (HCC), where PLIN5 expression is proven to be upregulated in hepatic tissue. Considering that HCC development is highly driven by cytokines present throughout NAFLD development and in the tumor microenvironment, we here explore the possible regulation of PLIN5 by cytokines known to be involved in HCC and NAFLD progression. We demonstrate that PLIN5 expression is strongly induced by interleukin-6 (IL-6) in a dose- and time-dependent manner in Hep3B cells. Moreover, IL-6-dependent PLIN5 upregulation is mediated by the JAK/STAT3 signaling pathway, which can be blocked by transforming growth factor-β (TGF-β) and tumor necrosis factor-α (TNF-α). Furthermore, IL-6-mediated PLIN5 upregulation changes when IL-6 trans-signaling is stimulated through the addition of soluble IL-6R. In sum, this study sheds light on lipid-independent regulation of PLIN5 expression in the liver, making PLIN5 a crucial target for NAFLD-induced HCC.

## 1. Introduction

Hepatocellular carcinoma (HCC) is the most common type of primary liver cancer (90%) and the fourth leading cause of cancer-related death worldwide [1]. Viral hepatitis and alcohol consumption are crucial risk factors for HCC. However, due to sedentary lifestyle, non-alcoholic fatty liver disease (NAFLD) is rapidly becoming a dominant underlying cause of HCC [2]. Due to the fact that mortality rates run along with its incidence, understanding the molecular mechanisms of HCC development and progression is imperative in developing effective targeted therapies to combat this deadly disease [3]. As for any type of cancer, the detection of early stages diagnostic markers or targetable proteins, which are important for its progression, would be highly beneficial.

Perilipin 5 (PLIN5) has been studied in the context of NAFLD for several years. It belongs to the Perilipin/Adipophilin/TIP47 (PAT) family comprising five molecules that are lipid droplet-associated proteins regulating fat storage and lipolysis [4]. PLIN5 is highly expressed in oxidative tissues such as heart muscle and liver, where it is regulated by the peroxisome proliferator-activated receptor (PPAR) transcription factor family [5,6,7]. It has been shown that it acts as a regulator of lipolysis and fatty acid oxidation, thereby maintaining cellular lipid balance [8]. PLIN5 blocks lipolysis when cells are overloaded with lipids through its interactions with either adipose triglyceride lipase (ATGL) or its co-activator lipid droplet-binding protein comparative gene identification-58 (CGI-58) [9]. On the other hand, PLIN5 facilitates fatty acid flux from lipid droplets to mitochondria, thereby providing a physical link between the organelles during starvation. PLIN5 is also able to promote the transcription of target genes involved in mitochondrial biogenesis and oxidative metabolism through transcriptional complexes with Sirtuin 1 (SIRT1) and PPARγ co-activator 1α (PGC-1α) [10].

Since its discovery and expression analysis, PLIN5 has been extensively studied in the context of NAFLD. However, in recent years, its role in HCC development has become apparent. We recently showed that PLIN5 is strongly expressed in tumors of human HCC patients [11]. The finding was further confirmed in mouse livers in which HCC was genetically or experimentally induced by treatment with the genotoxic agent diethyl nitrosamine. These findings suggest the “lipid-independent” regulation of PLIN5 expression and a possible involvement in HCC development. A subsequent study connected HCC with a PLIN5-46808-AT splice variant, which might contribute to poor prognosis and metastasis in HCC, most likely under the regulation of aberrant Y-box protein 3 through the pathway of primary bile acid biosynthesis [12]. Moreover, we have shown that PLIN5 induces the pro-inflammatory response in steatotic livers, while promoting inflammasome activation via nuclear factor κ-light-chain-enhancer of activated B cells (NF-κB) in primary hepatocytes [13].

Considering that the HCC microenvironment is enriched with inflammatory mediators such as growth factors, cytokines and chemokines, it is important to understand how they drive HCC pathogenesis [14]. Prototypically, dysregulated transforming growth factor-β (TGF-β) signaling has already been proven to play central roles in inflammation, fibrogenesis, and immunomodulation in the HCC microenvironment [15]. Likewise, tumor necrosis factor-α (TNF-α) plays a central role in sorafenib resistance in HCC [16] and interleukin-1β (IL-1β) has been found to promote HCC metastasis [17]. Especially important in promoting HCC is interleukin-6 (IL-6), which is overexpressed in advanced stages of HCC [18]. IL-6 is a multifunctional inflammatory cytokine involved in anti-apoptosis, angiogenesis, proliferation, invasion, metastasis, and drug resistance. In addition, increased IL-6 expression in HCC patients is closely related to the reoccurrence of HCC and poorer prognosis [19]. In this regard, PLIN5 has been proven to regulate immune response, especially the cytokine release in NAFLD [20,21].

Based on these findings, which indicate a strong correlation of PLIN5 with various cytokines in the HCC pathology, we asked ourselves whether cytokines, which mediate the inflammatory response in HCC, regulate PLIN5 expression in some way. As no previous research has investigated the potential influence of cytokines on PLIN5 expression in NAFLD progression and HCC microenvironment, we studied the effects of several cytokines on PLIN5 expression in human liver cancer cell lines.

## 2. Results

### 2.1. Differential Cytokine Regulation of PLIN5 Expression

PLIN5 regulates cytokine expression in several liver pathologies including HCC [13,22]. However, it has not yet been reported that cytokines are able to regulate PLIN5 expression in liver cancer cell lines. Therefore, we initially investigated the effect of cytokines contributing to the HCC microenvironment on PLIN5 expression in the human liver cancer cell line Hep3B. We observed that stimulation with IL-6 for 24 h causes a strong upregulation of the PLIN5 protein expression in Hep3B cells (Figure 1A). However, none of the other cytokines tested (i.e., TNF-α, TGF-β, or IL-1β) were able to induce PLIN5 expression in these cells. In line with Western blot experiments, statistically significant *PLIN5* mRNA upregulation by IL-6 was confirmed by reverse transcription and quantitative real time-PCR (RT-qPCR) (Figure 1B). Interestingly, IL-6-mediated upregulation of PLIN5 expression was mitigated when IL-6 was applied in combination with TNF-α or TGF-β for 24 h, as confirmed by Western blot analysis (Figure 1C). Strongly increased PLIN5 protein levels were still present after simultaneous treatment with IL-6 and IL-1β, showing that IL-1β has no effect on IL-6-mediated PLIN5 upregulation. This effect was confirmed by RT-qPCR (Figure 1D). To deepen our understanding of the impact that combined cytokine and lipid treatment have on PLIN5 expression, Hep3B cells were subsequently treated with IL-6 in combination with either monounsaturated fatty acids (oleic and palmitic acid) or polyunsaturated fatty acids (arachidonic and linoleic acid). No synergistic effect on PLIN5 expression was observed due to the high PLIN5 upregulation caused by IL-6 alone (Appendix A).

### 2.2. PLIN5 Upregulation and IL-6R Expression Assessment in Different Liver Cancer Cell Lines

To investigate whether the effect of IL-6-induced PLIN5 upregulation is limited to Hep3B cells, human tumorigenic Huh7 and HepG2 cells were subjected to parallel analysis. Western blot analysis showed that Huh7 and HepG2 express PLIN5 (Figure 2A). The relative quantities of PLIN5 protein in the different cell lines were in line with RT-qPCR results in which Hep3B cells showed slightly higher levels of *PLIN5* mRNA expression than Huh7, while HepG2 cells had significantly lower levels.

We next comparatively determined the quantities of interleukin-6 receptor (IL-6R) and its soluble form (sIL-6R), comprising the extracellular portion of the receptor, in cell extracts or cell supernatants of the three cell lines by Western blot analysis. This analysis revealed that all cell lines expressed similar levels of intracellular and secreted soluble IL-6R that both have high affinity for IL-6 [23,24], while the expression of the transmembrane IL-6R showed significant differences (Figure 2B). In particular, HepG2 cells showed higher IL-6R expression than Hep3B and Huh7 cells, both on protein and at the mRNA level.

Comparison of PLIN5 levels between the three cell lines after treatment with IL-6 for 24 h revealed that the highest PLIN5 induction was provoked in Hep3B cells (Figure 2C). Data obtained at the protein level were consistent with the RT-qPCR results as *PLIN5* mRNA was significantly upregulated after stimulation with IL-6 for 24 h. However, mRNA levels of *PLIN5* in HepG2 were not significantly enhanced when treated with IL-6 (Figure 2D).

### 2.3. IL-6 Induced PLIN5 Upregulation Is Dose- and Time-Dependent

In the following, Hep3B cells were used for all further experiments, considering that Hep3B cells showed the highest PLIN5 upregulation upon IL-6 stimulation.

Western blot analysis showed that the induction of PLIN5 expression by IL-6 is dose- and time-dependent (Figure 3A,B). PLIN5 expression was similar when cells are treated with a concentration of IL-6 ranging from 2.5 to 15 ng/mL. However, the highest concentration of IL-6 (20 ng/mL) was able to further increase PLIN5 protein levels. Regarding the time dependence, PLIN5 expression was highest after 12 h of IL-6 treatment, while PLIN5 expression remained constant thereafter (Figure 3C), highlighting the importance of prolonged IL-6 signaling to induce PLIN5 expression. To further elucidate the localization of the PLIN5 protein, we next performed nuclear extraction and immunofluorescence staining. Both techniques demonstrated that PLIN5 has a cytosolic localization (Figure 3D,E).

### 2.4. IL-6 Signaling Is Mediated by Different Downstream Effectors

To elucidate the mechanism by which IL-6 mediates PLIN5 expression, the activation of known IL-6 downstream signaling pathways was investigated after IL-6 (10 ng/mL) stimulation for 15, 30, 45, 60 min and 24 h.

One common mediator of IL-6 signaling is the signal transducer and activator of transcription 3 (STAT3) [25,26]. In our experiments, STAT3 phosphorylation was observed as early as 15 min after stimulation with IL-6, and persisted for up to 24 h (Figure 4).

Therefore, we presumed that STAT3 mediates IL-6-induced PLIN5 expression. The second effector that showed similar behavior was STAT1, which was activated after 30 min of IL-6 treatment. At the same time, the protein kinase B (PKB), commonly known as AKT and a downstream target of phosphoinositide 3-kinases (PI3K), was activated by IL-6 after 24 h.

The third pathway activated by IL-6 was the extracellular-signal-regulated kinase (ERK) pathway, which was activated for a short period upon IL-6 stimulation. Its activation was highest after 30 min stimulation and decreased gradually towards the 24 h mark. Once again, PLIN5 upregulation was confirmed by Western blot analysis after 24 h of IL-6 treatment.

### 2.5. JAK-Mediated IL-6 Signaling Is Involved in Regulation of PLIN5 Expression

To further elucidate the mechanism by which IL-6 induces PLIN5 expression, we next used different small molecule inhibitors. Ruxolitinib is a common inhibitor of Janus kinase (JAK) phosphorylation, blocking phosphorylation of STAT3, STAT1, AKT, and ERK [27]. Initially, we optimized the concentration of Ruxolitinib by determining the maximum concentration levels that do not cause apoptosis in the Hep3B cells using Puromycin (5 μg/mL), a compound well known to cause apoptosis, as a positive control [28]. None of the Ruxolitinib concentrations used induced apoptosis as indicated by the lack of increase in quantities of cleaved poly (ADP-ribose) polymerase (PARP) and caspase-3 (CASP3) in Western blot analysis (Figure 5A).

Next, the cells were pretreated with 5 μM Ruxolitinib for 1 h prior to 24 h stimulation with IL-6 (10 ng/mL). The inhibitory effect of Ruxolitinib was proven through blocked STAT phosphorylation (Figure 5B). Western blot analysis showed reduced PLIN5 expression after pretreatment with Ruxolitinib. At the same time, PLIN5 expression was strongly induced by treatment with IL-6 and accompanied by strong STAT3 phosphorylation. In line with this, significantly increased *PLIN5* expression was also confirmed at the mRNA level by RT-qPCR (Figure 5C). Taken together, these findings suggest that IL-6-induced PLIN5 expression in Hep3B cells is mediated by JAK kinase as an early step in signal transduction.

### 2.6. STAT3-Mediated IL-6 Signaling Is Involved in Regulation of PLIN5 Expression

To identify more specific downstream mediators of IL-6 signaling, we next used the selective STAT3 inhibitor STATTIC. This small molecule inhibitor inhibits activation, dimerization, and nuclear translocation of STAT3 [29]. We first optimized the concentration of STATTIC to avoid apoptosis in the Hep3B cells using Puromycin-treated cells as positive control. None of the tested concentrations of STATTIC (i.e., 2.5, 5.0, and 10 µM) induced apoptosis as tested by expression analysis for cleaved PARP and CASP3 in Western blot, while the treatment with Puromycin (5 μg/mL) resulted in cleavage of PARP and CASP3 (Figure 6A). Therefore, the cells were pretreated with 10 μM STATTIC for 1 h prior to 24 h stimulation with IL-6 (10 ng/mL) in the next set of experiments. Western blot analysis showed that STATTIC blocked IL-6-induced STAT3 phosphorylation and reduced PLIN5 expression (Figure 6B).

IL-6 induced PLIN5 upregulation was accompanied by strong STAT3 phosphorylation. In line with these findings, *PLIN5* mRNA levels were significantly increased by IL-6 and blocked by pretreatment with STATTIC as demonstrated by RT-qPCR (Figure 6C). Therefore, we concluded that the induction of PLIN5 expression by IL-6 requires STAT3 phosphorylation in Hep3B cells.

### 2.7. PI3K-Mediated IL-6 Signaling Is Involved in Regulation of PLIN5 Expression

Considering that AKT phosphorylation was enhanced by IL-6 stimulation after 24 h (cf. Figure 3), it was important to assess whether PI3K signaling has an impact on PLIN5 expression. LY294002 is a potent PI3K inhibitor, whose activity can be monitored by the reduction of AKT phosphorylation [30]. In a first set of experiments, we optimized the concentration of LY294002, showing that Hep3B cells tolerated concentrations of 5–10 µM well, while higher concentrations (i.e., 15 μM) of LY294002 induced apoptosis, as indicated in Western blot analysis by the increased quantities of cleaved PARP and CASP3 (Figure 7A).

We next pretreated Hep3B with 10 μM LY294002 for 1 h prior to its stimulation with IL-6 (10 ng/mL) for 24 h. In contrast to the inhibition with Ruxolitinib and STATTIC, Western blot analysis showed that the pretreatment of cells with LY294002 does not affect PLIN5 expression levels after 24 h (Figure 7B). However, in line with our previous experiments, we observed that PLIN5 expression is strongly induced by treatment with IL-6. The blockage of PI3K was confirmed via Western blot analysis through abolished AKT phosphorylation.

In summary, these findings suggest that PLIN5 induction caused by IL-6 is not mediated by PI3K in Hep3B cells.

### 2.8. TNF-α and TGF-β, but Not IL-1β Block IL-6-Mediated PLIN5 Upregulation

Cytokine signaling in obesity-associated NASH is strongly associated with liver cancer progression [31]. To further elucidate the reason behind the TNF-α- and TGF-β-mediated blockage of IL-6 induced PLIN5 upregulation (cf. Figure 1), we next treated Hep3B cells with IL-6 in combination with either TNF-α or TGF-β for 24 h. Interestingly, we found a strong reduction in STAT3 phosphorylation (Figure 8). On the other hand, IL-1β, which does not influence PLIN5 levels (cf. Figure 1A), does not interfere with IL-6 signaling through STAT3 phosphorylation. Consistent with previous data, the simultaneous stimulation with IL-6 and IL-1β resulted in a significant stimulation of lipocalin 2 (LCN2) expression, showing the biological functionality of these cytokines [32]. In addition, the stimulation with TGF-β provoked the expected phosphorylation of the respective downstream TGF-β mediators Smad2 and Smad3 [33,34].

At the mRNA level, the detection of IL-6 signaling components such as IL-6R and STAT3 showed great differences between the treatments in Hep3B cells. In particular, IL-6 stimulation in combination with IL-1β upregulates IL-6R subunit α (*IL6RA*) transcription, while TGF-β and TNF-α either reduce it or prevent its upregulation, respectively (Appendix A). *STAT3* mRNA levels are strongly upregulated by IL-6 treatment, however, both TGF-β and TNF-α prevent strong *STAT3* upregulation (Appendix A).

In sum, these experiments show that both, TNF-α and TGF-β, can effectively block IL-6-induced PLIN5 expression.

### 2.9. Classical and IL-6 Trans-Signaling Regulate PLIN5 Expression

We next focused on the impact of classical and IL-6 trans-signaling on PLIN5 expression. Both signaling pathways require IL-6, but their activity is somewhat different. While IL-6 can bind directly to IL-6R and provoke homodimerization of the signal transducer gp130, the complex of IL-6 and sIL-6R triggers dimerization of gp130, thereby inducing responses on cells that do not express membrane bound IL-6R [24]. In this set of experiments the three human liver cancer cell lines Hep3B, Huh7 and HepG2 were treated for 24 h with either IL-6 alone (15 ng/mL) or in combination with sIL-6R (100 ng/mL) to simulate classical or IL-6 trans-signaling. Downstream signaling was assessed through Western blot analysis of pSTAT3. Interestingly, a significant difference in STAT3 phosphorylation after stimulation with IL-6 and sIL-6R was detected only in Huh7 cells. However, the quantities of PLIN5 protein detected were not affected by the soluble IL-6 receptor (sIL-6R) in any of the cells, implying that PLIN5 expression is independent of IL-6 trans-signaling (Figure 9A). However, several differences were observed at the mRNA level. While there is no significant difference between the treatments in Hep3B cells, there was a decrease in *PLIN5* mRNA levels in Huh7 cells upon sIL-6R treatment (Figure 9B). This finding sheds light on the potentially differential PLIN5 regulation via classical/ trans-IL-6 signaling in Huh7 cells.

## 3. Discussion

Thus far, research on PLIN5 has focused mainly on its role in lipid homeostasis and NAFLD pathology [35,36]. Within that context, PLIN5 has been described as the key regulator in maintaining a balance between lipolysis and lipid oxidation depending on the conditions in cells [37]. PLIN5 ameliorates NAFLD, alleviates hepatic steatosis and fibrosis, and protects against hepatic injury in NAFLD [13].

It was only a few years ago that PLIN5 started to emerge as a potential prognostic biomarker or a treatable target in several types of cancer [11]. Nevertheless, the regulation of *PLIN5* expression in the HCC pathology is still not elucidated. Considering that HCC development is highly driven by cytokines present in the tumor microenvironment [38], we decided to explore the possibility of PLIN5 being regulated by cytokines commonly known to contribute to NAFLD and HCC pathologies. In particular, we investigated the impact of IL-1β, IL-6, TNF-α, and TGF-β on *PLIN5* expression in different human liver cancer cell lines.

We show for the first time that PLIN5 expression in liver cancer cells lines is regulated by cytokines. IL-6 caused a strong upregulation of *PLIN5* mRNA and protein in Hep3B cells. Potentially this effect can be explained by several other findings depicting IL-6 as a modulator of lipid metabolism in adipocytes, a stimulator of lipolysis and fat oxidation, and a regulator of liver metabolism during fasting [39,40,41]. Therefore, considering that PLIN5 fine-tunes lipid metabolism, it would be expected that PLIN5 is inducible by IL-6.

In order to assess whether the effect of IL-6 is limited to Hep3B, we next used Huh7 and HepG2 cells. We first compared the basal levels of PLIN5 and IL-6R among the different cell lines and found a great variation among cell lines in both PLIN5 and IL-6R expression. The Hep3B cell line showed the highest expression of PLIN5, while HepG2 showed the lowest on protein level, which was confirmed at the mRNA level. This is not surprising considering that they stem from different etiologies [42]. IL-6R is commonly found in hepatocytes [43], so we expected to detect its expression in all cell lines. The Huh7 cell line had the lowest expression of transmembrane IL-6R, potentially indicating its lower responsiveness to IL-6. Interestingly, upon IL-6 treatment, Huh7 cells, as well HepG2, showed upregulation of PLIN5 at the protein level regardless of IL-6R quantity. However, elevated *PLIN5* transcription was not statistically significant for HepG2. As expected, cells which do not express IL-6R, such as hepatic stellate cell derivatives (e.g., GRX and LX-2) showed no effect of IL-6 on *PLIN5* expression (Appendix A). Nevertheless, since we proved that IL-6-mediated upregulation of PLIN5 at the protein level is not limited to a specific liver cell line, we concluded that this is a newly discovered mechanism of PLIN5 regulation in hepatic cancer cells.

The effect of IL-6-mediated PLIN5 upregulation seemed to be both time- and concentration-dependent. This was expected based on previous reports on IL-6 responsiveness in other cell types and tissues [23,44]. Interestingly, elevated *PLIN5* expression was found to be dependent on the time of exposure to IL-6. This finding, obtained through removing the media with IL-6 after the indicated time, potentially depicts PLIN5 upregulation as a consequence mainly of prolonged or chronic exposure to IL-6. Considering that there are reports of both IL-6 and PLIN5 upregulation in HCC [11,45], it would be interesting to see whether this correlation is persistent and significant.

IL-6 acts by binding to the IL-6 receptor (IL-6R), causing gp130 dimerization and then triggering JAK phosphorylation and downstream signal activation [19]. To further elucidate which signaling axis/pathway mediates IL-6-induced PLIN5 upregulation, we analyzed the activation of common downstream signaling pathways after IL-6 treatment in Hep3B cells such as the JAK/STAT, PI3K/AKT and MEK/ERK pathways. We found JAK/STAT3/STAT1 and MEK/ERK pathway to be activated shortly after IL-6 stimulation. However, only the phosphorylation of STAT3 and STAT1 persisted for up to 24 h. As previously mentioned, PLIN5 expression is consistent with prolonged exposure to IL-6, suggesting that the JAK/STAT3 pathway is responsible for PLIN5 upregulation in Hep3B cells considering its prolonged activation.

To further explain the implication of respective pathways on *PLIN5* regulation, we tested the impact of different inhibitors on IL-6 treated Hep3B cells. Based on our observation that STAT3 was strongly phosphorylated after IL-6 addition, we assumed that the JAK/STAT3 pathway regulates PLIN5. In line with this hypothesis, the JAK inhibitor Ruxolitinib and the STAT3 phosphorylation inhibitor STATTIC both inhibited PLIN5 upregulation. This finding confirms previous reports showing PLIN5 being regulated by the JAK/STAT3 pathway after leptin treatment in adipocytes [46,47]. Therefore, we concluded that the JAK/STAT3 signaling axis mediates PLIN5 expression upon IL-6 stimulation in Hep3B.

Analysis of PI3K/AKT signaling revealed that AKT is phosphorylated after 24 h of IL-6 treatment. Considering that it has already been proven that the PI3K signaling pathway regulates PLIN5 expression in HepG2 cells [48], we decided to block PI3K using the common inhibitor LY294002. After making sure that the applied dose of LY294002 is non-toxic, PLIN5 expression was assessed. However, LY294002 was not able to fully abolish the IL-6-induced PLIN5 upregulation after 24 h, which gave rise to the conclusion that the PI3K/AKT pathway does not regulate PLIN5 expression in this setting. Nevertheless, it would be worth assessing whether AKT phosphorylation is actually the consequence of PLIN5 upregulation and not IL-6 signaling itself.

Interestingly, the effect of IL-6-mediated STAT3 phosphorylation and PLIN5 upregulation could be blocked by both TGF-β and TNF-α (Figure 8). Previous inhibitor treatments for JAK and STAT3 have confirmed that PLIN5 is upregulated due to JAK/STAT3 signaling. Therefore, we concluded that PLIN5 expression is blocked in that setting, at least in part, through STAT3 phosphorylation blockade.

Previous studies reported crosstalk of TGF-β and IL-6 in dampening IL-6 signaling by decreasing transmembrane IL-6 receptor expression [49], which is in line with our mRNA results. On the other hand, even though TNF-α does not reduce IL-6R levels as confirmed by RT-qPCR, it prevented its upregulation, thus leading to the same effect. However, the exact mechanism of how TGF-β and TNF-α mediate the reduction of IL-6 signaling and PLIN5 upregulation via decreased STAT3 phosphorylation remains to be elucidated.

IL-6 is known to trigger either pro- or anti-inflammatory responses depending on whether it induces canonical (classical) or trans-signaling [50]. Classical signaling through IL-6R is limited to cells expressing transmembrane IL-6R such as hepatocytes, monocytes, and leukocytes [51]. Trans-signaling is mediated by soluble IL-6R and enables IL-6 mediating signaling on all cells. Therefore, we assumed that PLIN5 expression could be differentially regulated by classical and trans-signaling. To test this hypothesis, different cell lines were treated with either IL-6 alone or in combination with commercially available recombinant sIL-6R. We found that STAT3 phosphorylation was substantially higher only in Huh7 cells, which is consistent with the finding that the Huh7 cell line has less transmembrane receptors than Hep3B and HepG2. Interestingly, the Huh7 cell line showed decreased levels of *PLIN5* mRNA due to the competition of classical and IL-6 trans-signaling, suggesting that PLIN5 expression is not only regulated by IL-6 but also depends on the ratio of IL-6R and sIL-6R. This was expected considering there had already been several reports referring to the complexity of IL-6 signaling [24,52].

Nevertheless, the exact effect of both classical and IL-6 trans-signaling on PLIN5 expression needs to be further elucidated in order to address the issue of PLIN5 being a result of an IL-6 pro- or anti- inflammatory response. It might be possible that the precise knowledge on the interactions of the different cytokines in the regulation *PLIN5* expression might offer new therapeutic opportunities to interfere with fatty liver diseases or its progression to HCC.

## 4. Materials and Methods

### 4.1. Cell Culture

The human cell lines Hep3B (#ACC93, Leibniz-Institute SMZ-German Collection of Microorganisms and Cell Cultures, Braunschweig, Germany), Huh7 (#01042712, Merck KGaA, Darmstadt, Germany) and HepG2 (#ATCC-HB-8065, LGC Standards GmbH, Wesel, Germany) are routinely used as a model to study liver cancer. HepG2 is an immortalized cell line originating from the liver tissue of a 15-year-old Caucasian male who had a well-differentiated HCC [41]. The Huh7 cell line, established in 1982 [53], originates from a liver tumor of a 57-year-old Japanese male and possesses p53 point mutation. The Hep3B cell line is derived from an 8-year-old black male suffering from HCC [41]. All cell lines were cultured in Dulbecco’s Modified Eagle Medium (DMEM) supplemented with 10% fetal bovine serum (FBS), 100 U/mL penicillin, 100 μg/mL streptomycin, 2 mM L-glutamine and 1 mM sodium pyruvate (all from Sigma-Aldrich, Taufkirchen, Germany) on 10 cm plates at 37 °C with 5% CO_2_ atmosphere and 95% humidity. Media were replaced every 48 to 72 h and cells were subcultured when 80% confluency was reached. For subculturing, either trypsin (#R001100, Gibco, ThermoFisher Scientific, Schwerte, Germany) for HepG2 or Accutase (#A1110501, Gibco, ThermoFisher Scientific) for Huh7 and Hep3B were used.

### 4.2. Cytokine and Inhibitor Stimulation

For stimulation experiments, the cells were seeded on 6-well plates and grown until 80% confluency was reached. The cells were then starved in DMEM containing 0.5% FBS for 16 h. Stimulations were completed in DMEM containing 0.2% FBS. For the treatments, 10 ng/mL of recombinant human TNF-α (#210-TA-005/CF, R&D Systems, Wiesbaden, Germany), 2.5 ng/mL of recombinant human TGF-β (#240-B-002/CF, R&D Systems), 2.5 ng/mL of recombinant human IL-1β (#HZ-1164, Miltenyi Biotech., Bergisch Gladbach, Germany) and indicated concentrations of IL-6 (Miltenyi Biotech.) were used for the indicated times. In order to analyze IL-6 trans-signaling, 100 ng/mL of soluble IL6-Rα human (sIL-6R, #SRP3097, Sigma-Aldrich) was used. To inhibit potential signaling pathways downstream of IL-6, cells were pretreated for 1 h with either JAK inhibitor Ruxolitinib (5 µM, #HY-50856), STAT3 inhibitor STATTIC (10 µM, # HY-13818) or PI3K inhibitior LY294002 (10 µM, #HY-10108), all obtained from MedChem Express (Hölzel Diagnostika Handels GmbH, Cologne, Germany). Puromycin-dihydrochloride was obtained from Sigma-Aldrich (#P8833). The fatty acids oleic acid (#O3008), palmitic acid (#P0500), arachidonic acid (#10931) and linoleic acid (#L1376) used for IL-6 co-stimulation experiments in Hep3B were all obtained from Sigma Aldrich and used at indicated concentrations.

### 4.3. Protein Analysis

Conditioned cell media were collected, centrifuged (4 °C, 10,000× *g*, 15 min) to remove residual cells and stored at −20 °C until use. For total protein isolation, RIPA buffer containing 20 mM Tris-HCl (pH 7.2), 150 mM NaCl, 2% (*w*/*v*) NP-40, 0.1% (*w*/*v*) SDS, and 0.5% (*w*/*v*) sodium deoxycholate supplemented with Complete ^TM^ inhibitor cocktail (#4693132001, Roche Diagnostics, Mannheim, Germany) and Phosphatase inhibitor cocktail 2 (#P5726, Sigma-Aldrich) was used. Protein concentration of whole cell lysates was determined using colorimetric DC protein assay (Bio-Rad Laboratories GmbH, Munich, Germany). Equal protein amounts (30 μg of whole cell lysates and 25 μL of supernatants) were prepared in Nu-PAGE LDS electrophoresis sample buffer (ThermoFisher Scientific) supplemented with dithiothreitol (DTT) as reducing agent (final concentration of DDT was 0.05 M). After denaturation at 95 °C for 5 min, samples were separated on 4–12% Bis-Tris gel (Invitrogen, ThermoFisher Scientific) and prepared in Nu-PAGE LDS electrophoresis sample buffer (ThermoFisher Scientific) with MES as running buffer. Proteins were electroblotted onto Amersham™ Protran^®^ nitrocellulose membranes (Merck, Darmstadt, Germany), with equal loading and successful transfer being confirmed by Ponceau S staining. Subsequently, non-specific binding sites were blocked in 5% (*w*/*v*) milk in TBST. Primary antibodies were diluted in 2.5% non-fat milk in TBST and applied overnight at 4 °C with shaking. The primary antibodies were visualized using secondary antibody coupled horseradish peroxidase (HRP) with the SuperSignal^TM^ chemiluminescent substrate (ThermoFisher Scientific). A detailed overview of the primary and secondary antibodies and expected protein sizes is provided in Appendix A. The observed molecular weight of target proteins was estimated from the SeeBlue^TM^ pre-stained protein standard (#LC5925, ThermoFisher Scientific) that was separated simultaneously in each gel with the samples.

### 4.4. RNA Isolation, cDNA Synthesis, and RT-qPCR

Total RNA was isolated from stimulated cells using RNeasy Mini kits (Qiagen, Hilden, Germany) according to the manufacturer’s instructions. First strand cDNA was synthesized from 1 μg of total RNA in 20 µL using SuperScript™ II reverse transcriptase and random hexamer primers (Invitrogen, ThermoFisher Scientific). For quantitative RT-PCR, cDNA, previously diluted 1:5 in sterile H_2_O, was amplified in 25 μL volume using SYBR^®^ GreenER™ qPCR SuperMix Universal for ABI PRISM^®^ (Invitrogen, ThermoFisher Scientific). All primers for RT-qPCR were purchased from Eurofins Genomics (Ebersberg, Germany). Primer sequences used in this study were designed using NCBI Primer Blast tool. Details about used primers are given in Appendix A.

### 4.5. Immunofluorescence Staining

Cells were seeded onto sterile, round glass coverslips. After reaching 80% confluence, cells were fixed in 4% paraformaldehyde for 15 min and gently washed with phosphate-buffered saline (PBS). Next, permeabilization with 0.2% Triton X-100 in PBS for 15 min at room temperature was performed. Blocking of non-specific parameters was completed in PBS supplemented with 0.1% Triton and 5% goat serum for 2 h, at room temperature. Primary antibody against PLIN5 (ProGen, #GP31, guinea pig) was diluted 1:200 in PBS and incubated overnight (4 °C, no shaking). For visualization, goat anti-rabbit cross-adsorbed Alexa Fluor 488 was applied at a dilution of 1:1000 in PBS and was incubated for 1 h at room temperature. Slides were subsequently mounted with anti-fade Fluoroshield Mounting Medium with DAPI (#104135, Abcam, Berlin, Germany) and were dried in the dark. The slides were stored at 4 °C until observation. Images were taken using Nikon Eclipse E80i fluorescence microscope and NIS-Elements Vis software version 3.22.01 (Nikon, Tokyo, Japan).

### 4.6. Nuclear Extraction

For nuclear extraction, cells were washed in PBS and transferred to hypotonic buffer (10 mM Hepes, pH 7.4, 10 mM KCl, 0.1 mM EDTA, 1 mM DTT, 0.5% NP-40 supplemented with Complete ^TM^ inhibitor cocktail) for 5 min on ice. After centrifugation (3000× *g*, 4 °C, 10 min), supernatant corresponding to cytosolic fraction was collected and stored to −20 until use. The pellet was washed in PBS, resuspended in second buffer (20 mM Hepes pH 7.4, 400 mM NaCl, 1 mM EDTA, 1 mM DTT supplemented with Complete ^TM^ inhibitor cocktail) and sonicated three times at 10 s intervals. Supernatant corresponding to nuclear fraction was harvested after centrifugation (15,000× *g*, 4 °C, 15 min). Equal protein amounts (20 μL of cytosolic fraction, 5 μL of nuclear fraction) were prepared in NuPAGE LDS electrophoresis sample buffer (ThermoFisher Scientific) supplemented with DTT as reducing agent.

### 4.7. Statistical Analysis

Data presented in this study are representative of the results of three independent experiments. Microsoft Excel was used for data processing, while statistical analysis was performed using GraphPad Prism software (version 8.0). For statistical analysis of RT-qPCR data, one-way analysis of variance (ANOVA) with Turkey’s multiple comparisons test was used. Probability value of *p* < 0.05 was considered as statistically significant. Results are shown as mean ± SD where differences between groups are marked as * *p* < 0.05; ** *p* < 0.01; *** *p* < 0.001.

## 5. Conclusions

In summary, we found a novel mechanism of *PLIN5* regulation in human liver cancer cells. Our results indicate for the first time a lipid-independent regulation of *PLIN5* expression via cytokines. Nevertheless, further studies addressing the impact of IL-6-mediated PLIN5 upregulation on its role in metastasis, apoptosis and proliferation need to be conducted.

## Figures and Tables

**Figure 1 ijms-24-07219-f001:**
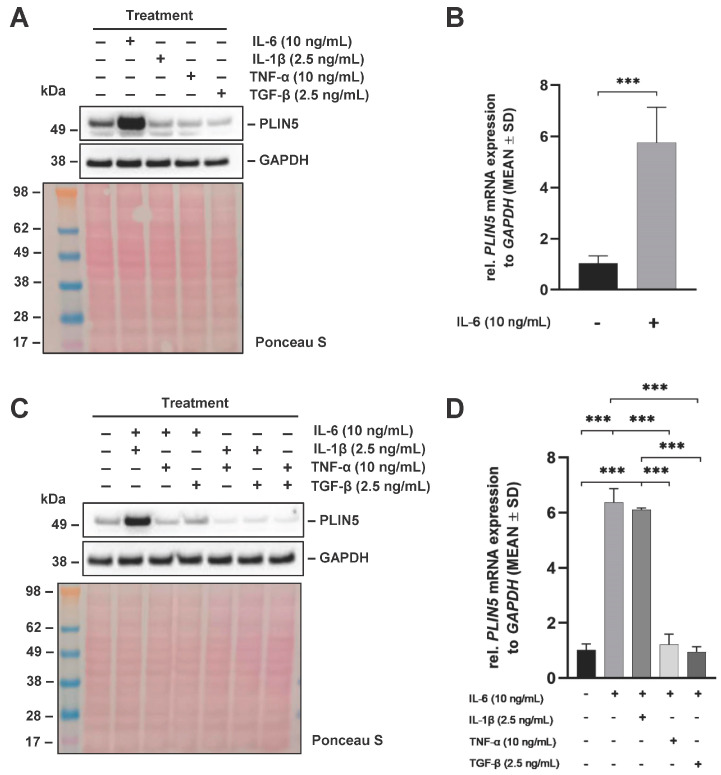
Differential cytokine regulation of PLIN5 expression in Hep3B cells. Hep3B cells were stimulated with the indicated concentrations of different cytokines and combinations thereof for 24 h. (**A**) Representative Western blot demonstrating induction of PLIN5 protein expression by IL-6 in Hep3B cells. GAPDH expression and Ponceau S staining served as controls to demonstrate equal protein loading. (**B**) Relative mRNA expression of *PLIN5* in Hep3B cells after stimulation with IL-6 (10 ng/mL) for 24 h provoked significant upregulation of *PLIN5* mRNA. (**C**) Western blot analysis of PLIN5 protein expression after stimulation with indicated cytokines and combinations thereof. (**D**) Relative *PLIN5* mRNA expression in Hep3B cells after stimulation with indicated cytokines and their combination for 24 h revealing that IL-6-mediated *PLIN5* upregulation is prevented by TNF-α and TGF-β, but not IL-1β. Data from RT-qPCR are shown as mean ± SD of three independent experiments performed in triplicate and measured in duplicate. Significant differences between the groups are indicated by asterisks (*** *p* < 0.001).

**Figure 2 ijms-24-07219-f002:**
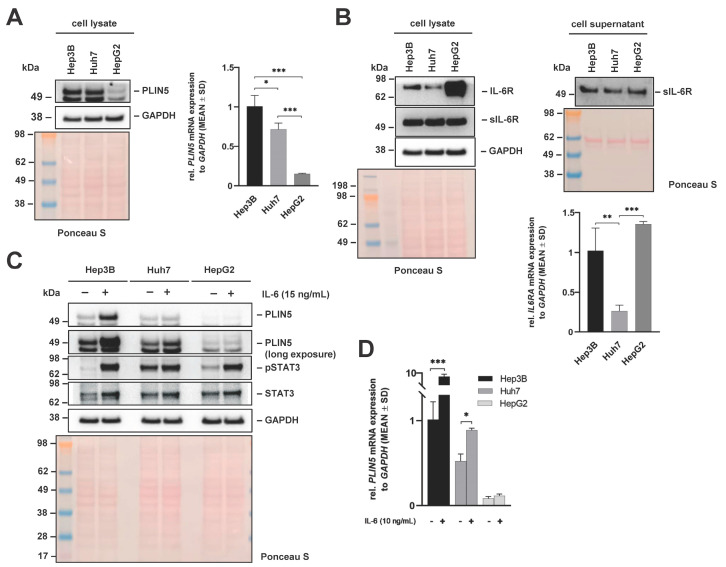
Basal expression of PLIN5, IL-6R, and sIL-6R after IL-6 stimulation in liver cancer cell lines. (**A**) Western blot of PLIN5 protein (left) and relative mRNA expression (right) in Hep3B, Huh7 and HepG2 cells reveal differences in PLIN5 expression between the different cell lines. (**B**) Western blot analysis of IL-6R and sIL-6R expression in cell extracts (left) and supernatants (middle) of Hep3B, Huh7 and HepG2 cells. In addition, the relative mRNA expression of *IL-6RA* in the three cell lines is depicted (right). Representative Western blot analysis (**C**) and RT-qPCR results (**D**) demonstrating induction of PLIN5 protein and relative mRNA expression in Hep3B, Huh, and HepG2 cells after treatment with IL-6 (10 ng/mL) for 24 h. For all Western blots, GAPDH detection and Ponceau S staining served as controls to demonstrate equal protein loading. Data for RT-qPCR results are shown as mean ± SD of three independent experiments performed in triplicate, measured in duplicate. Differences between the groups reaching significance are marked by asterisks (* *p* < 0.05, ** *p* < 0.01, *** *p* < 0.001).

**Figure 3 ijms-24-07219-f003:**
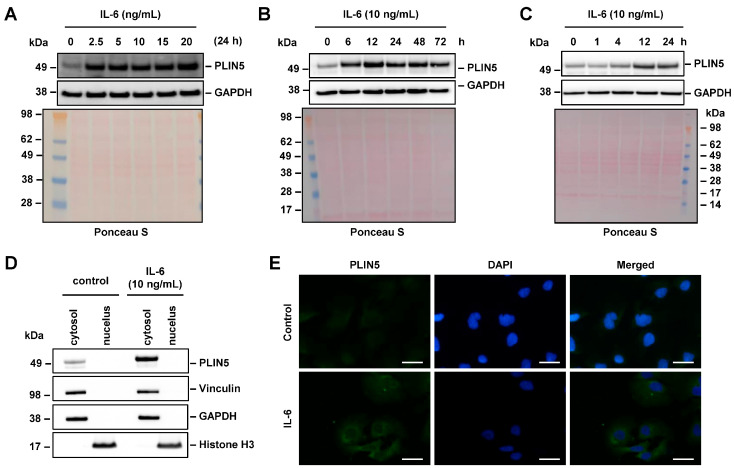
IL-6-mediated PLIN5 upregulation is dose- and time-dependent. (**A**) Hep3B cells were stimulated with rising IL-6 concentrations for 24 h and the cell extract probed for PLIN5 expression. (**B**,**C**) Hep3B cells were stimulated with IL-6 for the indicated time intervals and analyzed for expression of PLIN5 by Western blot analysis. (**D**) Western blot analysis depicting cytoplasmatic localization of PLIN5. For nuclear extraction, Histone H3 was used as control for the nuclear fraction, while the expression of Vinculin served as a marker for the cytoplasmic protein fraction. For all Western blots, GAPDH detection and Ponceau S staining served as controls to demonstrate equal protein loading. (**E**) Immunofluorescent staining of PLIN5 protein in Hep3B cell line after 24 h of IL-6 (10 ng/mL) stimulation confirmed cytoplasmic localization. Space bars represent 25 µm.

**Figure 4 ijms-24-07219-f004:**
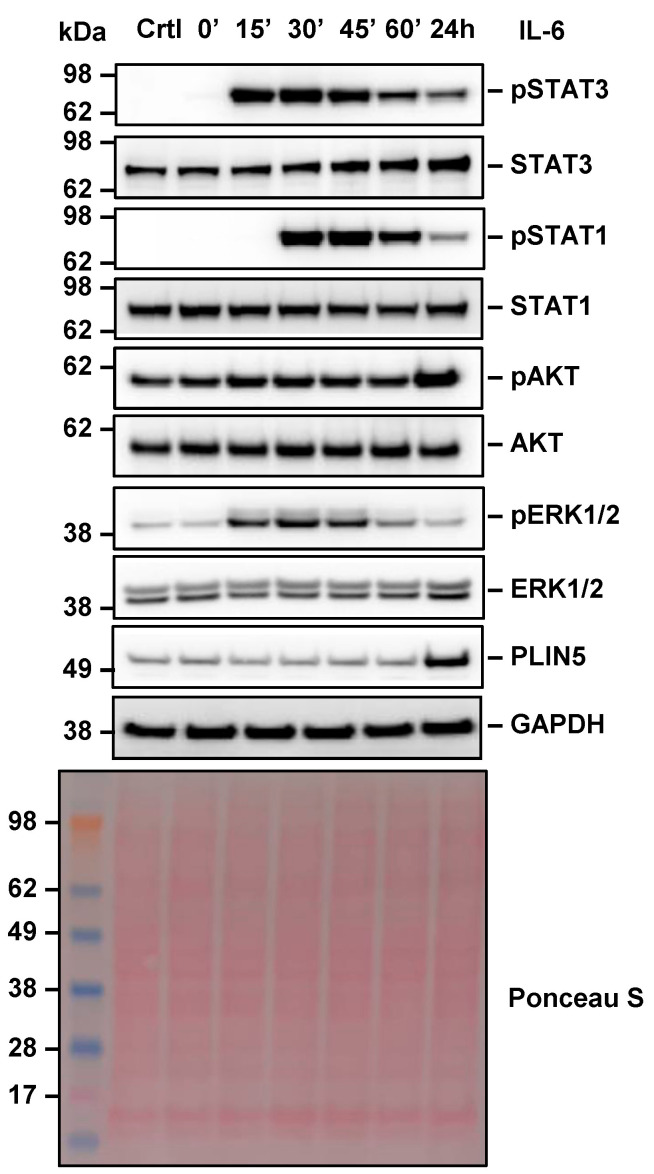
Effect of IL-6 stimulation on downstream signaling pathways in Hep3B cells. Hep3B cells were stimulated with IL-6 (10 ng/mL) for 0, 15, 30, 45, 60 min and 24 h and the activation of IL-6 downstream signaling components analyzed by Western blot analysis. GAPDH detection and Ponceau S staining served as controls to demonstrate equal protein loading.

**Figure 5 ijms-24-07219-f005:**
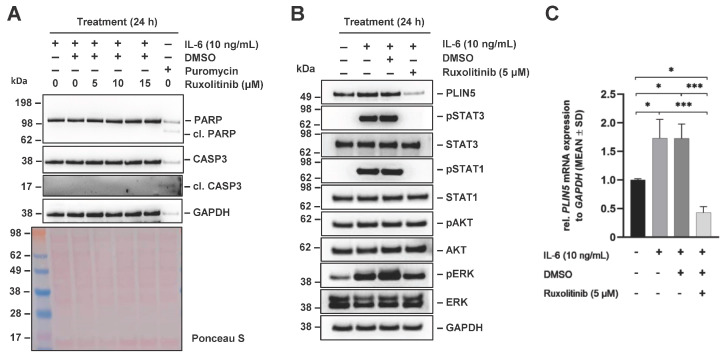
Effects of JAK inhibitor Ruxolitinib on IL-6-mediated PLIN5 upregulation in Hep3B cells. Hep3B cells were pretreated with the JAK inhibitor Ruxolitinib (5 µM) for 1 h and subsequently incubated with IL-6 (10 ng/mL) for 24 h. (**A**) None of the Ruxolitinib concentrations used induced apoptosis as demonstrated by the lack of elevated cleaved PARP or CASP3 quantities when compared with the apoptosis inducing control Puromycin (5 µg/mL). (**B**) PLIN5 expression and STAT3 phosphorylation, a downstream target of JAK, were analyzed by Western blot, demonstrating that Ruxolitinib blocked IL-6-induced PLIN5 protein expression. (**C**) Effect of Ruxolitinib and IL-6 treatment was assessed on *PLIN5* mRNA expression by RT-qPCR. GAPDH expression and Ponceau S staining served as controls to demonstrate equal protein loading. Significant differences between the groups are marked by asterisks (* *p* < 0.05, *** *p* < 0.001).

**Figure 6 ijms-24-07219-f006:**
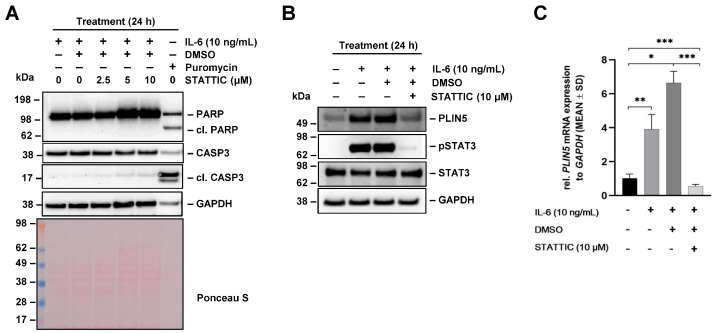
Effects of STATTIC on IL-6-mediated PLIN5 upregulation in Hep3B. (**A**) Hep3B cells were pretreated with STAT3 phosphorylation inhibitor STATTIC (10 µM) for 1 h and then treated for 24 h with IL-6 (10 ng/mL) and the expression of poly (ADP-ribose) polymerase (PARP), cleaved PARP, caspase-3 (CASP3), and cleaved CASP3 was analyzed by Western blot. (**B**) PLIN5 expression and STAT3 phosphorylation was analyzed by Western blot analysis demonstrating that STATTIC blocked IL-6-induced PLIN5 protein expression. The quantities of GAPDH and Ponceau S staining served as controls to demonstrate equal protein loading. (**C**) Effect of STATTIC on *PLIN5* mRNA expression was tested by RT-qPCR. Significant differences between the groups are marked by asterisks (* *p* < 0.05, ** *p* < 0.01, *** *p* < 0.001).

**Figure 7 ijms-24-07219-f007:**
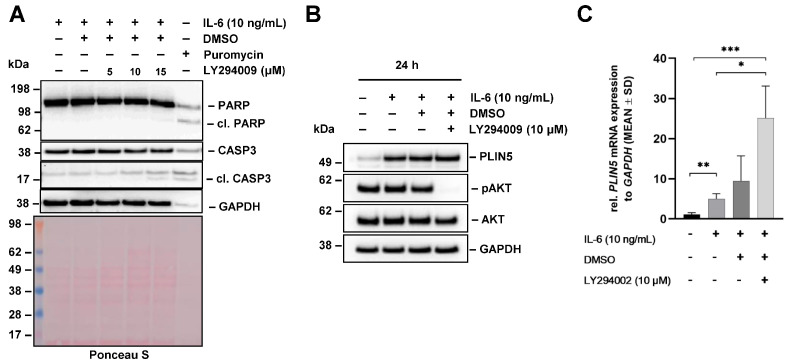
Effects of PI3K inhibitor LY294002 on IL-6-mediated PLIN5 upregulation in Hep3B cells. (**A**) Hep3B cells were pretreated with the indicated concentrations of PI3K inhibitor LY294002 for 1 h and then treated with IL-6 (10 ng/mL) for 24 h. Cell extracts were prepared and tested by Western blot analysis for PARP and cleaved caspase-3 (CASP3). Cell extracts prepared from cells that were treated with Puromycin served as control. GAPDH detection and Ponceau S staining served as controls to demonstrate equal protein loading. (**B**) PLIN5 expression and phosphorylation of AKT (downstream target of PI3K) analyzed by Western blot demonstrated that LY294002 has no influence on IL-6-induced PLIN5 protein expression. (**C**) Effect of LY294002 on *PLIN5* mRNA expression was tested by RT-qPCR. Significant differences between the groups are marked by asterisks (* *p* < 0.05, ** *p* < 0.01, *** *p* < 0.001).

**Figure 8 ijms-24-07219-f008:**
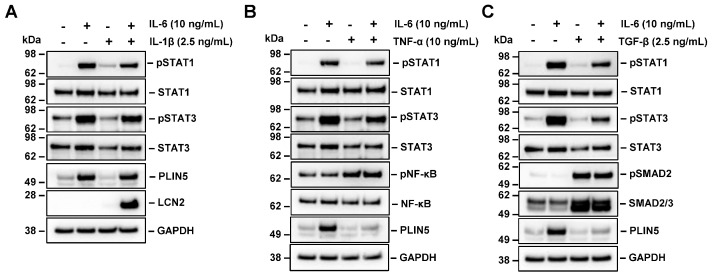
Effect of IL-1β, TNF-α and TGF-β on IL-6-mediated PLIN5 upregulation in Hep3B cells. (**A**–**C**) Hep3B cells were treated with IL-6 in combination with IL-1β, TNF-α and TGF-β for 24 h. PLIN5 expression and phosphorylation of STAT3 analyzed by Western blot demonstrated that (**A**) IL-1β has no effect on STAT3 signaling and PLIN5 expression while (**B**,**C**) TNF-α and TGF-β partially block STAT3 phosphorylation and IL-6-induced PLIN5 protein upregulation. GAPDH served as loading control.

**Figure 9 ijms-24-07219-f009:**
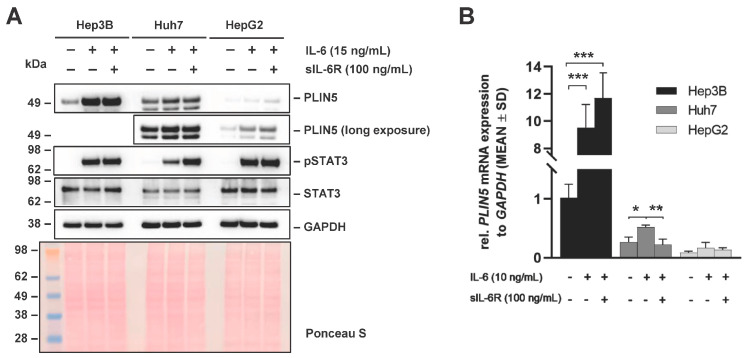
Effects on PLIN5 expression after simultaneous stimulation with IL-6 stimulation and sIL-6R in liver cancer cell lines. (**A**) Representative Western blot and (**B**) RT-qPCR results demonstrating the induction of PLIN5 protein and relative mRNA expression after IL-6 (15 ng/mL) stimulation alone or in combination with sIL-6R (100 ng/mL) for 24 h. GAPDH detection and Ponceau S staining served as controls to demonstrate equal protein loading. For RT-qPCR results, data are shown as mean ± SD of three independent experiments performed in triplicate and measured in technical replicates. Differences between the groups reaching significance are marked by asterisks (* *p* < 0.05, ** *p* < 0.01, *** *p* < 0.001).

## Data Availability

Additional data and repetitions of experiments depicted are available on request.

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
