# Peer review of "Lipid-Independent Regulation of PLIN5 via IL-6 through the JAK/STAT3 Axis in Hep3B Cells"

_ijms, 2023, doi:10.3390/ijms24087219_

Round 1

Reviewer 1 Report

Krizanac  et al have studied the regulation of PLIN5 expression by cytokines in Hepatocellular carcinoma cell lines. The author showed that PLIN5 expression is strongly upregulated by Interleukin-6(IL-6) in a concentration- and dose-dependent manner in Hep3B cells. Moreover, PLIN5 upregulation by IL-6 is mediated by the
JAK/STAT3 signaling pathway. I have the following suggestions for the authors. 

  • Authors claim that the mechanism they elucidate for IL6-mediated PLIN5 expression is lipid independent. However, there is no strong evidence provided in the manuscript. The provided data suggest that IL6 treatment upregulates PLIN5 expression, however, it does not indicate the source of IL6, given the interlink between IL6 and lipid metabolism. I find the statement confusing. Authors should clarify the same  
  • Assays (e.g., cell proliferation assays) showing the role of IL-6-mediated PLIN5 expression on liver carcinogenesis will help meet the paper's objective. 
  • Assays showing the synergistic effect of IL6 and lipids over PLIN5 expression could enhance the impact of the work as in cancer tissues multiple of these factors acts simultaneously on the cells to bring the carcinogenic changes and effects, rather than one at a time. 
  • A simple model diagram showing the signaling mechanism of regulation of PLIN5 expression can be useful to better understand the findings.

Author Response

Dear Reviewer 1,

many thanks for the time you spent in reading our manuscript. Please find our comments to your helpful suggestions in the attached pdf-file.

Regards

Ralf Weiskirchen

Reviewer 2 Report

The article "Lipid-Independent Regulation of PLIN5 via IL-6 through the 2 JAK/STAT3 Axis in Hep3B Cells" is devoted to the role of Perilipin 5 (PLIN5) - a crucial target for NAFLD-induced HCC.

The authors, using modern adequate research methods, studied the effects of several cytokines on PLIN5 expression in liver cancer cell lines. The data obtained by the authors expand our knowledge about the interactions of the different cytokines in regulation PLIN5 expression might offer new therapeutic opportunities to interfere with fatty liver diseases or its progression to HCC.

The study was carried out at a high scientific level, the interpretation of the results was carried out using modern scientific literature.

As a recommendation, I would advise the authors to use ANOVA with Turkey 's multiple comparisons tesn instead of Student's t-test when comparing the results of several experimental groups.

Author Response

Dear Reviewer 2,

many thanks for the time you spent in reading our manuscript. Please find our comments to your helpful suggestions in the attached pdf-file.

Regards

Ralf Weiskirchen
